# Drug Repurposing with a Graph-of-Thoughts Inspired Reasoning Framework

## Abstract

Drug repurposing is a promising strategy to accelerate therapeutic development, and provides a viable and effective way to treat diseases, especially rare diseases, that otherwise do not have established and approved treatment options. As the magnitude of available biological data continues to increase, computational methods have become vital for extracting meaningful insights and identifying candidates for repurposing. Although large language model (LLM)-driven agentic workflows show potential, their computational cost and latency often make them impractical. We present a novel, efficient platform that addresses this challenge by grounding LLM reasoning in a biomedical knowledge graph (PrimeKG). This uses known relationships between biological entities to deduce relevant drug repurposing information. Our method first identifies multiple diverse paths between a given drug-disease pair from a natural language query. A Graph-of-Thoughts (GoT)-inspired module then constrains the LLM to reason over these structured paths and synthesize the information into a coherent biological hypothesis. Our platform presents key novelties such as Graph-of-Thoughts-inspired reasoning and diverse KG path-finding that seek to ground reasoning in biological knowledge and mitigate hallucinations. Our evaluation demonstrates that this constrained approach performs comparably in accuracy to unconstrained agentic workflows. Our platform achieves this with significantly fewer LLM calls, 55.3% lower token consumption, and 40.4% less time. This GoT-inspired framework, grounded on knowledge graph data, presents an efficient and powerful system for LLM-driven drug repurposing, effectively balancing reasoning with computational efficiency.

## 1 Introduction

The traditional drug discovery process is a complex and costly endeavor, often spanning a decade or more, with costs reaching billions of dollars (Deore et al., 2019; Shareef et al., 2024). It typically involves going through several phases, some of which include target selection, safety review, clinical research and development, and FDA approval (Hughes et al., 2011). Thus, drug repurposing, identifying new uses for existing drugs, is emerging as a promising solution, providing a path forward to shorter timelines and a reduction in high rates of clinical failure (Cha et al., 2018). Drug repurposing utilizes existing drugs with established clinical efficacy and safety profiles, allowing drug candidates to bypass initial drug development research and safety evaluation phases (Pushpakom et al., 2019; Saranraj & Kiran, 2025), thereby saving time and reducing costs. Furthermore, the use of drugs with established clinical efficacy and safety profiles mitigates risks and reduces failures typically associated with traditional drug discovery (Parvathaneni et al., 2019). By reducing costs, time, and clinical failures, drug repurposing holds the promise for great societal impact as it leads to faster, more cost-effective treatments for diseases. Furthermore, it provides a path forward for treating rare diseases that otherwise do not have established and approved treatment options (Roessler et al., 2021; Jonker et al., 2024).

There are generally four phases involved in repurposing an existing drug for a new indication. First, the drug repurposing candidate must be identified based on initial research on pathways, mechanisms of action, safety, and efficacy, among other factors (Kulkarni et al., 2023). This phase can take around 1-2 years. Then, the proper license for the drug must be obtained, which can take 0-2 years. The drug development phase can then begin, which involves preclinical and clinical research to

evaluate the drug for treatment. This phase may span anywhere from 1 to 6 years. Finally, when the drug is publicly available, post-market safety surveys are carried out to monitor drug safety (Xue et al., 2018). This process significantly speeds up the process of developing therapeutics and provides expanded options for conditions that lack or have limited treatments.

Information from biomedical databases, existing clinical trials, reports of off-label uses, and other published data are often used to identify and select drug candidates. However, as the volume of biomedical data has grown exponentially, computational approaches have become essential for understanding and reasoning through this data (Park, 2019; Jarallah et al., 2025). Central to these computational methods is the use of Knowledge Graphs (KGs), large-scale graph networks that structure information by representing entities like drugs, proteins, and diseases as nodes, and their relationships as edges (Perdomo-Quinteiro & Belmonte-Hernández, 2024). One such KG is PrimeKG (Chandak et al., 2023), an expansive KG that integrates several data sources to map relationships between biological entities. The graph structure is ideal for uncovering hidden connections, forming the basis for identifying potential repurposing candidates.

## 1.1 BUILDING BLOCKS

Although such data is readily available, interpreting it requires sophisticated reasoning. This makes Large Language Models (LLMs) a valuable tool. In addition to text generation capabilities, LLMs can be used as powerful reasoning engines (Sahoo et al., 2025). LLM reasoning frameworks leverage prompting techniques to guide models through multi-step problem-solving processes (Fagbohun et al., 2024), enabling them to analyze evidence, follow logical sequences, and synthesize information to form complex conclusions.

The foundational step in the evolution of prompting techniques was *Chain-of-Thought* (Wei et al., 2022), which improves upon existing LLM reasoning capabilities by instructing them to break down their thought process into smaller steps. This allows the LLM to sequentially reason through smaller tasks that lead to the overall answer, simulating a thought process similar to that of a human.

While powerful, a single reasoning chain can be prone to errors. To address this, the concept was extended to Multiple CoTs, or *Chain of Thought with Self-Consistency (CoT-SC)* (Wang et al., 2022), which builds upon CoT by generating multiple reasoning chains and selecting the most consistent answer among them, further solidifying the reasoning process.

The CoT and CoT-SC prompting techniques still largely follow linear reasoning paths. However, a more advanced approach, *Tree of Thoughts (ToT)* (Yao et al., 2023a), introduced a more flexible structure. This prompting technique enables an LLM to explore multiple reasoning paths like a decision tree, allowing it to consider several thought chains in parallel, evaluate intermediate steps, and backtrack from mistakes. This process builds upon previous prompting techniques to provide more robust solutions.

The *Graph-of-Thoughts (GoT)* (Yao et al., 2023b) framework represents a significant evolution in prompting techniques. Moving beyond the linear nature of CoT, GoT formalizes the reasoning process itself as a graph. With GoT, individual reasoning steps or ideas are nodes, and the LLM can generate, evaluate, and aggregate these thoughts in a flexible, non-linear structure. This allows for the exploration of multiple reasoning paths, the aggregation of convergent lines of evidence, and the removal of unpromising ones. While GoT has shown promise in general problem-solving and creative tasks, its application in specialized scientific domains like biomedicine remains largely unexplored.

The most advanced applications of these reasoning capabilities are found in complex agentic LLM workflows (Derouiche et al., 2025; Hosseini & Seilani, 2025; Sapkota et al., 2025; Acharya et al., 2025). These systems can autonomously interact with tools, search databases, and synthesize findings, effectively following a similar process to that of a human researcher. While powerful, their unconstrained, iterative nature leads to a high volume of API calls and extensive token consumption, making them prohibitively slow and expensive for the kind of high-throughput screening required to systematically evaluate thousands of drug-disease pairs.

To address this gap, we propose a novel platform that combines the structured knowledge of a biomedical KG with the analytical power of a GoT-inspired reasoning framework. Our core thesis is that by constraining the LLM to generate thoughts along diverse biological paths, found in a

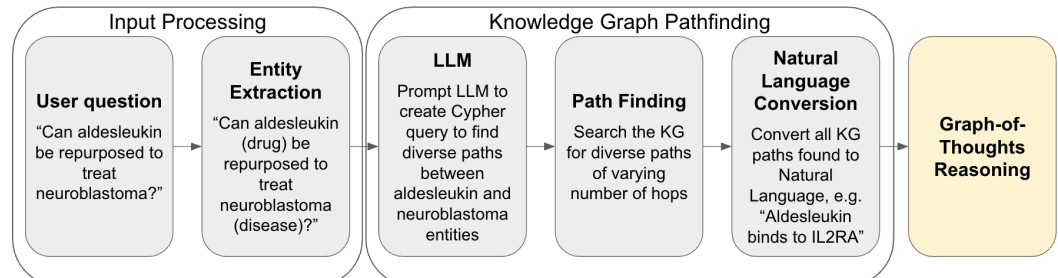

Figure 1: Architecture Overview. Given a drug repurposing question with a drug and disease, the drug and disease entities are extracted from the natural language question to add context. An LLM is then prompted to create a Cypher query for path finding. The query is executed and all returned paths are converted to natural language as context for the LLM to be used in the Graph-of-Thoughts reasoning process.

biomedical knowledge graph, our platform generates high-quality repurposing hypotheses that are comparable to those from resource-intensive agentic workflows, but with a lower computational cost.

In this paper, we present a novel application of a Graph-of-Thoughts-inspired framework to drug repurposing, grounding LLM reasoning in a real-world biomedical knowledge graph. We provide a quantitative and qualitative analysis, demonstrating that our method achieves comparable results to a baseline agentic workflow with significantly greater computational efficiency.

## 2 METHODS

### 2.1 SYSTEM ARCHITECTURE OVERVIEW

Our drug repurposing platform is designed as a multi-stage pipeline that integrates natural language processing (NLP), knowledge graph traversal, and advanced LLM-based reasoning. It was designed with the purpose of transforming a simple, natural language question into a well-supported hypothesis. The entire workflow is implemented within a Jupyter Notebook environment, orchestrating the interactions between the NLP model, the graph database, and the reasoning module. The process can be broken down into three key stages, as shown in Figure 1: Input Processing, Knowledge Graph Pathfinding, and GoT Reasoning. Several prompts and the natural language conversion structure throughout the workflow were adapted from K-paths (Abdullahi et al., 2025), a framework that extracts paths from dense biomedical KGs for drug repurposing and interaction tasks.

### 2.2 INPUT PROCESSING AND ENTITY RECOGNITION

The workflow begins with the user's query (e.g., "Can Aldesleukin be repurposed to treat neuroblastoma?"). To add context to this query, we use a specialized biomedical NLP model, en_ner_bc5cdr_md, from scispaCy (Neumann et al., 2019) for named entity recognition (NER). To enhance the model's accuracy and ensure it correctly identifies the specific entities present in our knowledge graph, we augment its capabilities using an Entity Ruler. This component is pre-loaded with lists of all drug and disease names, extracted directly from PrimeKG. This step ensures that when the user's question is processed, the specific drug and disease entities are accurately identified and tagged as a drug, protein, or disease, converting the unstructured text into a contextualized input for the next stage.

### 2.3 KNOWLEDGE GRAPH PATHFINDING IN PRIMEKG

With the drug and disease entities identified, the system queries the PrimeKG database to find potential mechanisms of action.

### 2.3.1 DATA SOURCE

The foundation of our pathfinding module is PrimeKG, a comprehensive biomedical knowledge graph comprising 129,375 nodes and over 4 million relationships. We selected PrimeKG for its extensive coverage of diverse biological entities and its integration of multiple data sources. Furthermore, PrimeKG was built to be extensible and easily updated from literature. This allows it to be constantly updated as new research becomes available, making it an ideal knowledge base for our platform. The graph's nodes encompass a wide range of types, including molecular entities (e.g., gene/protein, drug), clinical concepts (e.g., disease, effect/phenotype), and biological contexts (e.g., biological_process, pathway). The relationships between these nodes are similarly diverse, capturing interactions from the molecular level (e.g., protein_protein, drug_protein) to the clinical (e.g., indication, contraindication). To store and query this large-scale graph, we utilized Neo4j AuraDB, a managed graph database optimized for complex traversal and pathfinding operations (Guia et al., 2017).

### 2.3.2 PATH DIVERSITY

A core objective of our methodology was to find multiple mechanistically diverse pathways between a drug and a disease, rather than only identifying the shortest paths. Discovering varied paths is crucial, as it can reveal different potential mechanisms of action, suggest pharmacological effects, or provide convergent evidence that strengthens a repurposing hypothesis. To achieve this, we implemented a multi-step Cypher query strategy that leverages Neo4j's apoc.path.expandConfig procedure with a Depth First Search (DFS) algorithm in order to find paths of varying lengths in a memory-efficient manner. The process is designed to efficiently find meaningful paths by first identifying the direct neighbors of the target disease node and then searching for paths from the source drug node to this collection of neighbors. A key innovation in our approach is the dynamic generation of the database query. An LLM (Gemini 2.5 Flash) is prompted to construct a Cypher query specifically designed to find multiple, diverse paths between the drug and disease nodes in PrimeKG. This approach is more flexible than using static queries and is tailored to prioritize varied and mechanistically interesting connections.

The path expansion was carefully constrained using several key parameters within the procedure:

Path Length: We specified a minimum and maximum number of hops for the expansion from the drug to the disease's neighbors. This ensures the resulting paths are long enough to be mechanistically interesting while avoiding overly convoluted connections.

Uniqueness: The uniqueness constraint was set to NODE_GLOBAL, which ensures that no node is traversed more than once within any given path. This is critical for preventing cyclical or redundant paths and ensuring each result represents a unique sequence of biological events.

Relationship Filtering: To focus the search on the most biologically relevant interactions, a curated set of relationship types was selected. This selection prioritizes core pharmacological and physiological connections, such as drug-protein interactions, disease-protein interactions, drug effects, and others. The selection was made by the LLM generating the Cypher query.

This tailored approach allows us to efficiently extract a set of diverse and high-quality paths to be analyzed in the reasoning module without overwhelming the amount of context available. Providing this information allows for reasoning over known facts directly from the KG, which helps mitigate hallucinations and promote mechanistic, rigorous reasoning. The following shows an example of a Cypher query incorporating this logic.

```
// Step 1: Find start, end, and the end node's neighbors
CALL db.index.fulltext.queryNodes('drugNames', 'Aldesleukin~') YIELD node
    AS drugNode LIMIT 1
CALL db.index.fulltext.queryNodes('diseaseNames', 'Neuroblastoma~') YIELD
    node AS diseaseNode LIMIT 1

MATCH (neighbor)-[]->(diseaseNode)
WITH drugNode, diseaseNode, COLLECT(neighbor) AS neighbors
```

```
// Step 2: Run path expansion to the neighbors
CALL apoc.path.expandConfig(drugNode, {
    minLevel: 4,
    maxLevel: 6,
    uniqueness: 'NODE_GLOBAL',
    terminatorNodes: neighbors,
    filterStartNode: false,
    relationshipFilter: 'drug_protein>|protein_protein>|disease_protein>|
    drug_effect>|phenotype_protein>'
}) YIELD path

// Step 3: Match the final step and return a map representing the path
WITH path, diseaseNode
MATCH (lastNodeInPath)-[lastRel]->(diseaseNode) WHERE lastNodeInPath =
    nodes(path)[-1]
RETURN {nodes: nodes(path) + diseaseNode, relationships: relationships(
    path) + lastRel} AS p
LIMIT 30
```

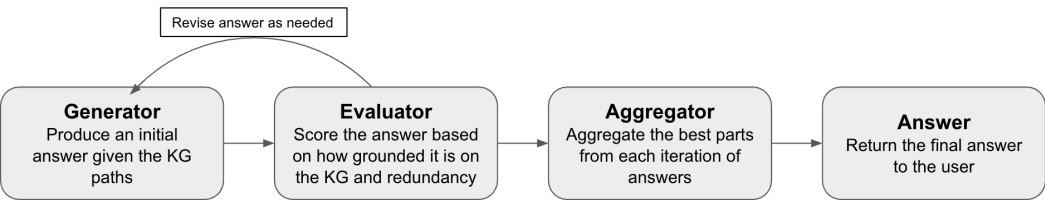

Figure 2: GoT Reasoning Overview. The Generator node serves as the entry point for reasoning by generating an initial answer. The answer is then evaluated and if needed, iteratively refined until the score meets the threshold or reaches the revisions threshold. The strengths of each answer is then aggregated into a cohesive final answer and returned to the user.

## 2.4 THE GRAPH-OF-THOUGHTS (GOT) REASONING MODULE

To systematically evaluate the biological plausibility of the pathways identified from PrimeKG, we developed a reasoning module inspired by the Graph-of-Thoughts (GoT) framework (Besta et al., 2024). GoT structures LLM reasoning as a graph, allowing for the generation, evaluation, and synthesis of multiple intermediate "thoughts" in a non-linear fashion. Our implementation formalizes this process into a directed state graph to analyze each KG path, ensuring that the final output is a well-reasoned and well-supported mechanistic hypothesis. The workflow was constructed using the LangGraph library and consists of four primary nodes: Generation, Evaluation, Aggregation, and Answer, as shown in Figure 2.

The process begins at the *Generation node*, which serves as the entry point for the reasoning work-flow. This node's function is to produce an initial hypothesis, or "thought," that explains the mechanistic connection between the drug and the disease. For context, the LLM receives all KG paths, translated from its graph structure into a natural language description. Given this context, the LLM is prompted to generate a preliminary answer explaining how the drug could treat the disease. It is also asked to explain its reasoning, provide the exact KG triples it used in its answer, and a simple "yes" or "no" for evaluating accuracy.

Once an initial hypothesis is generated, it is passed to the *Evaluation node* for assessment. This node employs a separate LLM-based evaluator to score the hypothesis against two key criteria, ensuring both factual accuracy and informational value:

Grounding Score (0-10): This metric assesses how well the generated text adheres to the information provided in the KG path context. A high score indicates that the reasoning is directly supported by the provided data, minimizing the potential for model hallucination.

Redundancy Score (0-10): This metric evaluates the novelty and conciseness of the information presented. It penalizes responses that are overly repetitive or fail to synthesize the path information into a coherent insight.

These two scores are averaged to produce a final quality score. This evaluation step is critical, acting as a filter to ensure only high-quality, data-supported thoughts proceed. If a hypothesis achieves a final score of 9.0 or higher, it is considered validated and is passed to the Aggregation node. Otherwise, If the score is below 9.0, the hypothesis is routed back to the Generation node for improvement. This creates an iterative refinement loop where the model is explicitly guided on how to improve its reasoning. To ensure computational efficiency and prevent infinite loops, this refinement process is limited to a maximum of three iterations. If a satisfactory score is not achieved after three attempts, the last generated hypothesis is carried forward.

All validated hypotheses from the refinement loop are collected at the *Aggregation node*. At this node, a final LLM call synthesizes the key insights from each of the previous responses into a final, robust hypothesis. This step ensures that the strengths of each previous response are combined. Finally, the workflow terminates at the *Answer node*, which formats and returns this aggregated explanation as the final output for the drug-disease query.

---

**Algorithm 1** Workflow Process

---

**Require:** drug, disease
 1: Set question, formatted as "Can {drug} be repurposed to treat {disease}?"
 2: $relationships \leftarrow$ [List of all relationship types]
 3: $nodes \leftarrow$ [List of all node types]
 4: Set LLM prompt to generate Cypher query
 5: Use Gemini-2.5 Flash to generate Cypher query
 6: Run the query and get triples formatted in a DataFrame
 7: Convert triples to natural language
 8: **if** Paths were found in the KG **then**:
 9:     $final\_answer \leftarrow$ Run Graph-of-Thought reasoning on the paths
10: **else**
11:     $final\_answer \leftarrow$ "Answer: Searched knowledge graph for up to 6 hops and no paths
12:     were found between {drug} and {disease}. Cannot determine drug repurposing value"
13: **end if**
14: **return** final_answer

---

This process is advantageous because it allows the LLM to revise its work, as opposed to a zero-shot setting where the first answer produced is final. This results in a more cohesive and robust final answer.

## 2.5 BASELINE AGENTIC WORKFLOW FOR COMPARISON

To contextualize the performance of our constrained reasoning pipeline, we established a baseline using Biomni (Huang et al., 2025), a state-of-the-art, multi-purpose biomedical AI agent. Biomni represents a fundamentally different approach to hypothesis generation, operating within a vast, unified agentic environment that integrates over 150 specialized tools, 59 databases, and 105 software packages. This architecture allows it to autonomously select and execute a wide range of actions to answer complex biomedical queries.

For the comparison, we posed the exact same set of natural language questions to Biomni that were used to evaluate our system (structured as, "Can Drug X be repurposed to treat Disease Y?"). Unlike our pipeline, which is constrained to a specific knowledge graph, Biomni was free to leverage all accessible tools. Observational analysis of its workflow revealed that its primary strategy involved deploying literature search tools to query biomedical publications. It also utilized predictive models, such as the TxGNN graph neural network, and performed computational simulations, including docking studies, to formulate its conclusions. By benchmarking against Biomni, we can directly compare our focused, knowledge graph-driven methodology with a powerful, unconstrained agentic system. This comparison allows us to evaluate the trade-offs between computational efficiency, reasoning transparency, and predictive accuracy inherent in each approach.

# 3 RESULTS

## 3.1 QUANTITATIVE EVALUATION AND BENCHMARK COMPARISON

To rigorously assess the performance and unique strengths of our drug repurposing pipeline, we conducted a comprehensive quantitative evaluation. The core of this analysis was performed on a subset of the PharmaDB dataset, which was previously utilized in the K-Paths study. This subset comprises 93 drug-disease pairs characterized by a concentrated focus, evaluating multiple candidate drugs against a smaller set of disease indications. Additionally, we constructed a custom evaluation dataset comprising 93 unique drug-disease pairs. This dataset was carefully curated to include a balanced mix of positive cases (drugs known to be effective for the given disease) and negative cases (drugs not indicated for the disease), providing a robust dataset for evaluating predictive accuracy. The custom dataset provides a wider subset of diseases to further evaluate the generalizability of our pipeline.

Each drug-disease pair was formatted as a natural language query ("Can {drug} be repurposed to treat {disease}?") and processed through the entire pipeline: knowledge graph pathfinding, followed by Graph-of-Thoughts (GoT) reasoning.

A key challenge in biomedical reasoning is the inherent sparsity of the underlying knowledge base. For 11 of the 93 pairs, the pathfinding component could not establish any connecting paths within PrimeKG. In these cases, the system cannot generate a prediction due to knowledge graph incompleteness and defaults to "no evidence, cannot conclude." These 11 pairs were therefore omitted from the quantitative analysis, resulting in a final evaluation set of 82 pairs.

In the context of early-stage drug repurposing, our evaluation strategy prioritizes a high recall score to ensure the minimization of false negatives. Although this decision entails a trade-off with precision (resulting in a higher rate of false positives), this is a desirable characteristic for initial screening. A false negative in this phase equates to prematurely discarding a potentially highly effective therapeutic candidate. However, in real-world drug development, the candidates identified in this initial in-silico stage would proceed to subsequent, resource-intensive experimental and clinical validation. These later, more rigorous steps are designed to effectively screen out the false positives.

Therefore, our primary objective is to maximize the capture of all plausible candidates (high recall) during the first pass, preventing the loss of any viable repurposing opportunity.

Table 1: GoT(ours) and Biomni Accuracy Metrics

| System | Dataset | Precision | Recall | F1-Score | Model |
|---|---|---|---|---|---|
| Biomni | PharmaDB | **0.66** | **0.86** | **0.75** | Gemini 2.5 Pro |
| GoT(ours) | PharmaDB | 0.57 | 0.66 | 0.61 | Gemini 2.5 Pro |
| CoT | PharmaDB | 0.41 | 0.65 | 0.50 | Gemini 2.5 Pro |
| Zero-shot | PharmaDB | 0.47 | 0.71 | 0.56 | Gemini 2.5 Pro |
| TxGNN | PharmaDB | 0.68 | 0.65 | 0.67 | N/A |
| Biomni | PharmaDB | **0.88** | 0.55 | **0.68** | GPT-4o |
| GoT(ours) | PharmaDB | 0.54 | **0.89** | 0.67 | GPT-4o |
| Biomni | Custom Dataset | 0.64 | 0.97 | 0.77 | Gemini 2.5 Pro |
| GoT(ours) | Custom Dataset | 0.75 | 0.89 | **0.81** | GPT-4o |

Table 2: Time and Token Consumption Metrics

| System | Mean Time per Query (seconds) ↓ | Mean Token Usage ↓ |
|---|---|---|
| Biomni | 114 | 121636 |
| GoT(ours) | 68 | 17280 |

The performance of our GoT-based pipeline on the PharmaDB subset is summarized in Table 1. The quantitative evaluation highlights the distinct advantages of our constrained, GoT-based methodology. Our system achieved a substantially higher recall (0.89) compared to Biomni (0.55) with the GPT-4o model. This indicates that when our pipeline proposes a repurposing candidate, it exhibits a high true positive rate, thereby minimizing false negatives. In a real-world drug screening scenario,

this high recall is vital as it ensures that viable repurposing candidates are not prematurely screened out or disregarded during the initial high-throughput analysis.

Biomni achieved a higher precision (0.88 vs. 0.54) with the GPT-4o model. While this suggests greater certainty in its positive predictions, this comes at the expense of a higher false negative rate. The trade-off is particularly compelling given our system's architectural simplicity, which achieves this output with significantly greater computational efficiency. We argue that in an initial screening stage, a system that prioritizes finding all plausible candidates (high recall) is more beneficial, as subsequent, resource-intensive laboratory validation will filter out the true false positives.

We attribute the slightly lower precision on the PharmaDB dataset to its concentrated nature. When the underlying knowledge graph contains sparse information for a specific, recurring disease, our system's ability to generate unique and valid reasoning paths for multiple different drug candidates is constrained. This dependency on the KG's richness is a core feature of our design.

To further contextualize our performance, we benchmarked our approach against TxGNN, a state-of-the-art graph neural network method. A direct comparison presented a challenge regarding dataset coverage: Of the 93 initial PharmaDB pairs, only 36 were present in the prediction output generated by TxGNN.

To ensure a fair comparison, the evaluation set for TxGNN was limited to these 36 overlapping drug-disease pairs. A prediction score threshold of 0.5 was used to classify results (scores >= 0.5 as positive). On this common subset of 36 pairs, TxGNN achieved a precision of 0.68, recall of 0.65, and F1-score of 0.67.

Our GoT pipeline achieved comparable results on this subset (precision 0.57, recall 0.66, F1 0.61). Critically, however, our pipeline was capable of generating predictions for 82 pairs, demonstrating superior generalizability to novel drug-disease combinations not explicitly optimized by the TxGNN training process. Furthermore, our approach provides transparent, interpretable reasoning traces in contrast to the black-box nature of the GNN-based prediction.

These results are highly instructive. The strong overall performance, particularly the high recall on the generalized dataset, demonstrates the system's effectiveness and robustness across a wide spectrum of repurposing questions. The PharmaDB benchmark further reveals that the system maintains efficacy comparable to much larger, more complex agentic or purely data-driven frameworks, underscoring the power of our structured, knowledge-grounded methodology to deliver reliable scientific hypotheses.

We attribute Biomni's higher precision largely to its reliance on literature-based methods for evidence aggregation. Since the PharmaDB evaluation dataset is constructed from established drug-disease pairs with confirmed indications, a literature-focused approach is naturally predisposed to high accuracy on existing, well-documented relationships. However, this reliance poses a limitation when evaluating novel drugs or exploratory candidates with little to no existing literature. Since our pipeline constrains the LLM to explicitly reason over paths retrieved from the underlying biomedical KG, this design allows for a thorough analysis of implicit connections between a drug and a disease, even if the connections have not been explicitly curated or summarized in the medical literature. Furthermore, by grounding the reasoning in structured knowledge (PrimeKG) rather than unstructured text summarization, our method is more applicable to evaluating novel drugs or underexplored relationships that are present in biology but may be missed by purely literature-based approaches.

## 3.2 QUALITATIVE ANALYSIS

### 3.2.1 CASE STUDY: REPURPOSING OF DACTINOMYCIN FOR THE REPURPOSING OF PERIPHERAL NERVOUS SYSTEM NEOPLASMS

In order to validate our results, we present a comparison, shown in Table 3, of an answer and reasoning provided by our GoT system and Biomni for the repurposing of Dactinomycin for peripheral nervous system neoplasms.

As shown in Table 3, the two systems used different information to come to their final conclusions. Biomni did not identify connections between Dactinomycin and proteins such as SUZ12, NF1, CDKN2A, and PNPLA6. However, SUZ12 and NF1 proteins are frequently inactivated MP-NSTs, accelerating tumor growth (Rad & Tee, 2016). Thus, it is important to consider this drug for

Table 3: Comparison of GoT and Biomni for question: Can Dactinomycin be repurposed to treat peripheral nervous system neoplasms?

|  | **GoT(ours)** | **Biomni** |
|---|---|---|
| Query | Can Dactinomycin be repurposed to treat peripheral nervous system neoplasms? | Can Dactinomycin be repurposed to treat peripheral nervous system neoplasms? |
| System Answer | Dactinomycin can be repurposed to treat peripheral nervous system neoplasms. | Dactinomycin is not a promising candidate for repurposing to treat peripheral nervous system neoplasms. |
| Reasoning | Dactinomycin has been shown to interact with ABCB1, a key transporter protein involved in drug efflux, which can significantly influence the pharmacokinetics of various drugs, including chemotherapeutics. The interaction of Dactinomycin with ABCB1 suggests that it may modulate the transport and bioavailability of other molecules involved in the pathogenesis of peripheral nervous system neoplasms. The downstream connections to UBC and proteins such as SUZ12, NF1, CDKN2A, and PNPLA6 indicate pathways that could be influenced by Dactinomycin, as these proteins are implicated in tumorigenesis and cellular regulation within the peripheral nervous system. | Dactinomycin functions as a non-specific cytotoxic agent by binding to DNA and inhibiting RNA synthesis. This is a general mechanism effective against many rapidly proliferating cells. However, peripheral nervous system neoplasms, such as malignant peripheral nerve sheath tumors (MPNSTs), are known to be driven by specific molecular alterations, primarily in the Ras/MAPK and PI3K/Akt/mTOR signaling pathways. Dactinomycin does not specifically target these pathways, making it a less-than-ideal candidate compared to modern targeted therapies.The TxGNN model, a powerful tool for predicting drug repurposing opportunities, did not identify Dactinomycin as a top candidate for treating malignant peripheral nerve sheath tumors.While Dactinomycin is approved for other sarcomas (e.g., Ewing sarcoma, rhabdomyosarcoma), its efficacy is context-dependent. |
| Correct Answer | Dactinomycin can be repurposed to treat peripheral nervous system neoplasms. | Dactinomycin can be repurposed to treat peripheral nervous system neoplasms. |

repurposing, which our system correctly identifies. This information is grounded in the paths found in PrimeKG and allows for more rigorous reasoning over information that may not be commonly highlighted in literature.

Dactinomycin has been studied for the treatment of MPNSTs in combination with other chemotherapy drugs (Martin et al., 2020). Although the improvement in survival benefits were marginal, it is still important to consider the drug for repurposing rather than rule it out immediately due to its interactions with highly relevant proteins involved in cell proliferation for MPNSTs.

# 4 DISCUSSION

## 4.1 THE POWER OF CONSTRAINED REASONING

The strong performance of our pipeline, particularly its high recall compared to a more complex agentic system, demonstrates the significant advantages of a constrained reasoning framework for scientific hypothesis generation. Our methodology is designed to limit the LLM's operational scope, yielding substantial benefits in terms of reliability, verifiability, and efficiency.

A major advantage of this approach is the mitigation of LLM hallucination. By providing the reasoning module exclusively with data in the form of KG paths, we anchor the reasoning process in a factual, structured knowledge base. This is critical in a biomedical context, where the accuracy of underlying data is vital (Pan et al., 2023). To enforce this, we prompt the LLM to cite the KG triples used to justify its conclusions. This allows researchers to easily verify the evidence supporting any generated hypothesis and ensures that the system's outputs are both transparent and trustworthy.

Furthermore, this constrained approach significantly enhances computational efficiency. By providing the LLM with a curated set of relevant facts, we drastically reduce its search space. Thus, the model's resources are entirely focused on synthesizing the data. This approach avoids the computational cost of exploratory steps common in unconstrained agentic workflows, leading to faster and more resource-efficient hypothesis generation.

The Graph-of-Thoughts (GoT) inspired reasoning framework is instrumental in elevating the quality of the final output. In a zero-shot or one-shot setting, an LLM generates a response in a single pass, with no opportunity for refinement. Our iterative process of generation, self-evaluation, and aggregation requires the model to critically assess its own reasoning. This allows it to identify and fix logical gaps to progressively build a more robust argument over multiple cycles. This results in a refined and cohesive answer that is more likely to represent a biologically plausible mechanism.

Finally, this approach prioritizes high recall to ensure that viable drug repurposing candidates are not prematurely discarded in early screening phases. This mitigates the risk of overlooking potentially highly effective treatments. Furthermore, the reliance on the pipeline's underlying KG to provide the LLM with relevant biological paths between drug and disease nodes not only serves to mitigate LLM hallucinations but additionally prioritizes reasoning over connections between biological nodes. This capability allows the LLM to effectively analyze and derive conclusions from intricate connections present in the KG structure but not yet explicitly aggregated or summarized in the medical literature. Consequently, our pipeline is especially promising for the exploratory evaluation of drug-disease pairs involving novel or largely unexplored pharmacological agents.

## 5 CONCLUSION

In this work, we presented a novel drug repurposing platform that effectively addresses the challenge of analyzing vast biomedical data. We demonstrate that a constrained, knowledge-graph-grounded reasoning framework can efficiently evaluate drug repurposing hypotheses. By first identifying diverse mechanistic paths in PrimeKG and then constraining a Graph-of-Thoughts (GoT)-inspired LLM module to reason exclusively over this evidence, we successfully produced hypotheses with sound, auditable analytical reasoning.

Our evaluation revealed that this structured approach achieves accuracy comparable to a sophisticated, unconstrained biomedical agent while significantly reducing computational overhead. The results demonstrate the power of grounding LLM reasoning in structured knowledge, which not only mitigates hallucinations but also focuses computational resources on synthesizing complex information. This work serves as a proof-of-concept for a new paradigm in computational scientific discovery: one that strategically combines the pattern-recognition capabilities of LLMs with the factual rigor of knowledge graphs. Such structured reasoning frameworks hold great potential to accelerate discovery not only in drug repurposing but across other scientific domains where data is abundant but requires computational methods for reasoning. Future work could address limitations such as being constrained to only KG information by exploring hybrid models. For example, a "human-in-the-loop" system could allow researchers to manually add or suggest alternative reasoning paths.

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

# A APPENDIX

## A.1 CASE STUDY: VALIDATING A KNOWN REPURPOSING OF ALDESLEUKIN FOR NEUROBLASTOMA

To validate our platform's ability to identify and reason through biologically relevant mechanisms, we tested it on the use of Aldesleukin for neuroblastoma treatment. Aldesleukin is indicated for the treatment of metastatic renal cell carcinoma and metastatic melanoma. However, it has also been

studied for neuroblastoma and has been approved to be used in a multi-drug regimen as part of the first-line therapy for pediatric high-risk neuroblastoma.

Query: Can Aldesleukin be repurposed to treat neuroblastoma?

Knowledge Graph Pathfinding Results: The system identified 30 distinct paths in PrimeKG connecting Aldesleukin to neuroblastoma. The top three paths, representing diverse potential connections, were:

Path 1: Aldesleukin binds to IL2RA. IL2RA interacts with GATA3. Neuroblastoma is associated with GATA3.

Path 2: Aldesleukin is metabolized or inhibited by PLA2G4A. PLA2G4A interacts with CTNNB1. Neuroblastoma is associated with CTNNB1.

Path 5: Aldesleukin is metabolized or inhibited by PLA2G4A. PLA2G4A interacts with UBC. UBC interacts with MYC. Neuroblastoma is associated with MYC.

### A.1.1   GoT Reasoning Module Output

Final Answer: Aldesleukin, also known as interleukin-2 (IL-2), has potential pathways that support its repurposing for neuroblastoma treatment based on its interactions within the knowledge graph. Aldesleukin binds to IL2RA (Interleukin-2 receptor alpha), which subsequently interacts with GATA3, a transcription factor involved in T cell differentiation and implicated in neuroblastoma. Additionally, Aldesleukin is metabolized or inhibited by PLA2G4A (phospholipase A2 group IVA), which interacts with key proteins associated with neuroblastoma, including CTNNB1 (beta-catenin), TP53 (tumor protein p53), and MYC (an oncogene). These interactions suggest that Aldesleukin's effects could modulate pathways relevant to neuroblastoma progression and survival, providing a basis for further investigation into its efficacy against this disease.

Reasoning: The reasoning is grounded in the interactions of Aldesleukin with IL2RA and PLA2G4A, which are linked to key oncogenic pathways and transcription factors associated with neuroblastoma. The binding of Aldesleukin to IL2RA activates GATA3, which is relevant to neuroblastoma biology, while the metabolic interactions with PLA2G4A influence proteins that are critical in neuroblastoma progression.

KG Triples Used:

- ('Aldesleukin', 'binds to', 'IL2RA')
- ('IL2RA', 'interacts with', 'GATA3')
- ('neuroblastoma', 'is associated with', 'GATA3')
- ('Aldesleukin', 'is metabolized or inhibited by', 'PLA2G4A')
- ('PLA2G4A', 'interacts with', 'CTNNB1')
- ('neuroblastoma', 'is associated with', 'CTNNB1')
- ('Aldesleukin', 'is metabolized or inhibited by', 'PLA2G4A')
- ('PLA2G4A', 'interacts with', 'TP53')
- ('neuroblastoma', 'is associated with', 'TP53')
- ('Aldesleukin', 'is metabolized or inhibited by', 'PLA2G4A')
- ('PLA2G4A', 'interacts with', 'MYC')
- ('neuroblastoma', 'is associated with', 'MYC')

### A.1.2   Biomni Answer

Aldesleukin, a recombinant form of interleukin-2 (IL-2), exerts its anti-tumor effects by stimulating the proliferation and activity of T-lymphocytes and Natural Killer (NK) cells. This general mechanism of enhancing the immune system's ability to recognize and eliminate cancer cells provides a broad rationale for its use in various cancers. Neuroblastoma is a pediatric cancer with several known molecular drivers, including MYCN amplification. While the tumor has its own specific genetic landscape, the principle of immune surveillance and the potential for an immune-mediated attack remains a valid therapeutic strategy. Clinical trials conducted in the early 1990s investigated the use of Aldesleukin in children with recurrent or refractory neuroblastoma. These studies demonstrated that while Aldesleukin had a manageable safety profile, its efficacy as a single agent was

limited, with only a small percentage of patients showing a partial response. Despite the modest results of these early trials, the fundamental mechanism of action of Aldesleukin remains relevant. The limited success could be attributed to various factors, including the specific patient population (recurrent/refractory disease), the dosing regimen, or the immunosuppressive tumor microenvironment of neuroblastoma. Modern immunotherapy approaches often involve combination therapies, where an agent like Aldesleukin could be used to augment the effects of other treatments like checkpoint inhibitors or CAR-T cell therapy. Therefore, there is a plausible, albeit not overwhelmingly strong, rationale for its repurposing, likely as part of a combination regimen.

### A.1.3  ANALYSIS OF RESPONSES

These results demonstrate the differences between our knowledge-graph grounded reasoning module and Biomni, a multi-purpose biomedical agent. The main differences in these answers are due to the underlying knowledge available. Biomni's tools were able to access to specific mechanistic information on Aldesleukin and clinical trial data, as well as relevant information on neuroblastoma. Our GoT-based module had access to the 30 KG paths that were found and successfully reasoned over the connections between the drug, genes/proteins, and the disease in a fraction of the time and compuational cost. While the provided hypothesis does not take into account the drug's safety and efficacy profiles, it provides a solid starting point for identifying drug repurposing candidates to be evaluated.

### A.2  LLM USE

Gemini 2.5 Pro was used to polish sections throughout this paper. It was used to clarify details that were unclear, strengthen language, formatting, and to connect disjoint ideas to improve cohesiveness.

