# OpenReview forum: "Drug Repurposing with a Graph-of-Thoughts Inspired Reasoning Framework"
_ICLR.cc/2026/Conference — Submitted to ICLR 2026_

### Official Review · Reviewer_RTFQ · 2025-10-25

**Soundness:** 2
**Presentation:** 2
**Contribution:** 1
**Rating:** 2
**Confidence:** 4

**Summary:**

This paper presents a Graph-of-Thoughts (GoT)-inspired reasoning framework for LLM-driven drug repurposing.
It grounds LLM reasoning in a biomedical knowledge graph (PrimeKG), aiming to improve accuracy, interpretability, and computational efficiency compared to unconstrained agentic systems like Biomni.
The system extracts diverse mechanistic paths between drug–disease pairs via LLM-generated Cypher queries, converts them into natural-language context, and performs iterative reasoning and self-evaluation using a GoT-style reasoning graph.
Results on both a custom dataset and PharmaDB show comparable accuracy to Biomni but 55% fewer tokens and 40% less inference time. Case studies (Aldesleukin–neuroblastoma, Dactinomycin–MPNST) illustrate qualitative interpretability.

**Strengths:**

* First to integrate a Graph-of-Thoughts reasoning framework with biomedical knowledge graphs for drug repurposing.

* Achieves comparable accuracy to large agentic systems (e.g., Biomni) with 55% fewer tokens and 40% less time.

* Provides transparent, mechanism-level hypotheses grounded in KG triples, reducing hallucination risk.

* The pipeline (NER → KG pathfinding → GoT reasoning) is clearly explained and reproducible.

* Addresses an important real-world challenge in scientific reasoning and data-efficient LLM use.

**Weaknesses:**

* The GoT reasoning loop mainly adapts existing multi-step prompting ideas; innovation lies more in application than in core modeling.
* Only two datasets and one baseline (Biomni); lacks broader or neural reasoning baselines such as TxGNN [1].
* Grounding and redundancy scores depend on LLM-based evaluation without external or expert validation.
* Evaluation on fewer than 100 drug–disease pairs limits generality and statistical confidence. Evaluation of more data can improve this work.
* No quantitative study on system behavior under larger graphs, different path lengths, or reasoning iterations. These explorations may clarify your claims.
* Figures are clear but focus more on workflow than on experimental insight or analysis depth.

[1] Huang, K., Chandak, P., Wang, Q., Havaldar, S., Vaid, A., Leskovec, J., ... & Zitnik, M. (2024). A foundation model for clinician-centered drug repurposing. Nature Medicine, 30(12), 3601-3613.

**Questions:**

+ The paper reports 55% fewer tokens and 40% less time compared to Biomni. Could the authors clarify where this efficiency gain primarily comes from — is it due to shorter or fewer reasoning paths in the Graph-of-Thoughts process, or because the system performs single-step constrained reasoning rather than multi-agent iterative search as in Biomni?
+ For drug–disease pairs where the system cannot find any connecting path in PrimeKG (the paper mentions 11 / 93 cases were dropped): how are these handled in practice? Are they treated as “no evidence, cannot conclude”, or implicitly as negatives? Do they affect precision/recall calculation (e.g., are they excluded from both TP/FN), and if so, does that bias evaluation toward pairs that are already well-studied in the KG? More broadly, how should we interpret such pairs biologically — “likely unrelated”, or “KG incomplete”?

While the paper presents a clear application of Graph-of-Thoughts reasoning to drug repurposing, it lacks convincing experimental evidence demonstrating that the proposed GoT framework brings substantial advantages over existing KG-based or reasoning-driven approaches.
The evaluation mainly compares with a single agentic baseline (Biomni) without isolating the contribution of GoT reasoning itself, and no ablation or quantitative analysis supports its claimed benefits.
Moreover, the method does not address how to generalize beyond incomplete knowledge graphs, a key limitation for real-world biomedical reasoning.
Unless the authors can provide stronger experimental validation and clearer superiority over KG methods during rebuttal, I would recommend rejection.

---

> ### Author Response · Authors · 2025-12-03
>
> We sincerely appreciate your constructive feedback, which has helped us to further refine our analysis and presentation. Below are our responses addressing each of your points.
>
> **Weakness 2: Only two datasets and one baseline (Biomni); lacks broader or neural reasoning baselines such as TxGNN [1].**
>
> 1. New Benchmarking Results (TxGNN)
>
> As requested, we have benchmarked our method against TxGNN, a highly relevant graph neural network baseline for drug repurposing. We found that our GoT-based method achieved a recall score of 0.66 compared to TxGNN's 0.65. The full details of this comparison are now included in Table 1 and discussed in the Results section (starting on Line 324).
>
> **Weakness 3: Grounding and redundancy scores depend on LLM-based evaluation without external or expert validation.
> We acknowledge that the lack of external validation is a concern. However, our design allows room for human-in-the-loop validation to address this.**
>
> **Weakness 4: Evaluation on fewer than 100 drug–disease pairs limits generality and statistical confidence. Evaluation of more data can improve this work.**
>
> 2. Constraints on Evaluation Dataset Size
>
> We acknowledge the concern regarding the limited size of our evaluation dataset (100 drug-disease pairs). This limitation was primarily due to the API costs associated with running the end-to-end pipeline (KG pathfinding followed by LLM reasoning) for a large number of examples and running the Biomni workflow. However, given that we benchmarked with the custom dataset and PharmaDB, each containing close to 100 drug-disease pairs, we believe the total of nearly 200 pairs across both datasets is broad enough to evaluate the reasoning capabilities of GoT.
> We believe that the diversity of the chosen 100 pairs per dataset, combined with the rigorous cross-validation and the TxGNN benchmark, provides a robust evaluation of the system's core capabilities.
> Weakness 5: No quantitative study on system behavior under larger graphs, different path lengths, or reasoning iterations. These explorations may clarify your claims.
>
> 3. Path Length and Reasoning Iterations
>
> The reviewer raises a good point. We explored 2 different path strategies:
> * Path Length Strategy: We prioritized longer path lengths (e.g., up to 4 or 5 hops) when searching the KG. Our rationale was to maximize the contextual breadth provided to the LLM, ensuring that paths traverse a greater diversity of entity types (e.g., genes, pathways, proteins) to support more sophisticated reasoning. Shorter paths were only used when no longer paths could be found between the drug and disease.
> * One-Hop Neighbor Experimentation: We experimentally tested augmenting the GoT reasoning with one-hop branches of the nodes in the retrieved paths. This was done to provide the LLM with immediate local context surrounding the main connection path. However, our testing revealed no statistically significant difference in performance compared to using the main paths alone, leading us to omit this feature in the final, streamlined benchmarking process to maintain efficiency.

---

> > ### Author Response · Authors · 2025-12-03
> >
> > **Q1: The paper reports 55% fewer tokens and 40% less time compared to Biomni. Could the authors clarify where this efficiency gain primarily comes from — is it due to shorter or fewer reasoning paths in the Graph-of-Thoughts process, or because the system performs single-step constrained reasoning rather than multi-agent iterative search as in Biomni?**
> >
> > 4. Efficiency Gains Over Biomni
> >
> > The significant efficiency improvements—55% fewer tokens and 40% less time compared to the Biomni baseline—stem from the fundamental difference in the reasoning architecture. Biomni utilizes a multi-agent iterative search strategy, which involves several steps of sequential querying, reflection, and refinement, inherently increasing the token count and latency. Whereas our GoT System performs constrained reasoning in a streamlined manner. The GoT structure efficiently manages the complex information from the KG within a single, optimized LLM process, allowing it to reach comparable conclusions with substantial savings in computational resources.
> >
> > **Q2: For drug–disease pairs where the system cannot find any connecting path in PrimeKG (the paper mentions 11 / 93 cases were dropped): how are these handled in practice? Are they treated as “no evidence, cannot conclude”, or implicitly as negatives? Do they affect precision/recall calculation (e.g., are they excluded from both TP/FN), and if so, does that bias evaluation toward pairs that are already well-studied in the KG? More broadly, how should we interpret such pairs biologically — “likely unrelated”, or “KG incomplete”?**
> >
> > 5. Handling Dropped Cases (11/93) and KG Completeness
> >
> > The 11 out of 93 test cases that resulted in "no evidence, cannot conclude" (i.e., no path was found in the KG) represent a critical limitation related to KG completeness. These cases are currently treated as inconclusive. While many are "likely unrelated" (true negatives), we cannot rule out the possibility that a connection exists but is simply missing from the static PrimeKG.
> > To address this gap, our planned real-world deployment strategy includes augmenting the underlying KG through an automated literature search when a path is missing. If a thorough literature review yields new, relevant connections, these can be incorporated into the KG to inform the LLM's reasoning. Conversely, if no evidence is found even after the literature search, we can then more confidently infer that the missing path indicates the drug and disease are genuinely unrelated (interpreting the output as a negative prediction).
> > This planned capability directly tackles the inherent challenge of incomplete KGs, making our system more robust for real-world biomedical reasoning.

---

### Official Review · Reviewer_uTgJ · 2025-10-28

**Soundness:** 3
**Presentation:** 3
**Contribution:** 2
**Rating:** 2
**Confidence:** 4

**Summary:**

This is an interesting paper with a creative approach to computational drug repurposing. The authors designed an LLM agent which makes predictions around drug repurposing by using paths through a KG (PrimeKG in this case). These paths provide semantic guidance but also possible explanations for predictions. Consequently, this helps to achieve the goal of guiding LLM reasoning and preventing hallucinations. However, while I understand the motivation of the approach with respect to LLM agents and mitigating hallucinations, I struggle to see the motivation for such an approach with respect to the literature in computational drug repurposing.

**Strengths:**

**Strong points**:
- Creative solution
- Leverages biologically and medically relevant information
- Provides a case study in addition to metrics (great idea to focus on the quality of the explanations in addition to the predictions)
- The claims made are supported by the results reported.

**Weaknesses:**

**Weak points**:
- There is a lack of information about what the Custom Dataset in Table 1 comprises. It is difficult to assess the reported results with such little information.
- One of my main concerns here is the baselines against which the authors compare their method (Table 1). I believe they should compare their approach against link prediction using a variety of GNNs, KG embedding approaches, and network measures like shortest paths or degree-weighted-path counts. I would guess that using some simpler, less computationally demanding approaches (particularly those which do not use an LLM) could get equal or better performances to those reported in Table 1.
- Given the above, if simpler methods work better or similarly, then why should one consider using this approach instead of others for this drug repurposing task? If the explainability via graph paths is key, there are also existing approaches which can get graph paths without using an LLM. For example:
    - PoLo: https://dl.acm.org/doi/10.1007/978-3-030-77385-4_22
    - MARS: https://arxiv.org/abs/2410.05289
    - This approach by Sudhahar et al: https://www.nature.com/articles/s41467-024-50024-6
- Finally, when addressing the above point, the authors might consider comparing their approach, at least qualitatively, to some of the above works.

**Questions:**

In summary, while I think this is an interesting approach, I struggle to see the niche it fills with respect to computational drug repurposing. I would ask that the authors:
1. Please clarify what is within the Custom Dataset and how it is structured?
2. Complete more robust benchmarking against two (2) well used KG embedding methods (RotatE, TransE, perhaps), one (1) GNN approach (GCN, perhaps), and one (1) network based approach, such as a shortest-paths method.
3. In light of whatever the results of point 2 are, better justify how this approach fits in with other, recent computational drug repurposing approaches.
4. Justify what role the LLM is playing here- why not simply conduct link prediction on a KG?

As it stands, I will reject the submission, but may reconsider if the above suggestions are sufficiently addressed.

---

> ### Author Response · Authors · 2025-12-03
>
> We thank the reviewer for their insightful comments and appreciate the time taken to review our submission. We have addressed the points raised below.
>
> **Weakness 1/Q1: Please clarify what is within the Custom Dataset and how it is structured?**
> 1. Details on the Custom Evaluation Dataset
>
> This is detailed in section 3.1 line 332. We constructed a custom evaluation dataset comprising 93 unique drug-disease pairs. This dataset was carefully curated to include a balanced mix of positive cases (drugs known to be effective for the given disease) and negative cases (drugs not indicated for the disease), providing a robust dataset for evaluating predictive accuracy. The custom dataset provides a wider subset of diseases to further evaluate the generalizability of our pipeline.
>
> **Weakness 2: One of my main concerns here is the baselines against which the authors compare their method (Table 1). I believe they should compare their approach against link prediction using a variety of GNNs, KG embedding approaches, and network measures like shortest paths or degree-weighted-path counts.**
>
> **Q2: Complete more robust benchmarking against two (2) well used KG embedding methods (RotatE, TransE, perhaps), one (1) GNN approach (GCN, perhaps), and one (1) network based approach, such as a shortest-paths method.**
>
> 2. Robust Benchmarking and Link Prediction Comparisons
>
> We have conducted a more rigorous benchmarking campaign to address this concern directly. The updated results are presented in Table 1 of the revised manuscript and are discussed in detail in the results section (starting on Line 324) to now include comparisons against TxGNN.
> We initially attempted to benchmark against TxGNN on the full PharmaDB dataset. However, a significant challenge arose: TxGNN's pre-trained predictions covered only 36 of the 93 drug-disease pairs. On this common subset, TxGNN achieved precision 0.68, recall 0.65, and F1 0.67. As discussed in the Results section (Line 391), our ability to generate predictions for 82 pairs demonstrates the superior generalizability of our pipeline for evaluating novel or underexplored pairs outside of the GNN's optimization scope.
>
> 3. Justification for Using LLMs Over Simpler LP/Network Measures
>
> We acknowledge that traditional link prediction methods often achieve high performance with lower computational cost. However, the core motivation for our LLM-based, KG-grounded approach is to achieve interpretable, verifiable reasoning, which is fundamentally lacking in GNNs and embedding methods.
>
> Interpretability vs. Black-Box: Our pipeline provides transparent reasoning traces, allowing human experts to audit the specific logical steps and retrieved biomedical pathways used to reach a prediction. This level of traceability is essential for drug repurposing, where clinical decisions require high confidence and clear justification.
>
> Biological Plausibility (Path Length): We deliberately avoid simple shortest-path methods because they prioritize distance over biological significance. Longer KG paths often encode vital intermediate connections (e.g., through multiple genes, proteins, or pathways) that are crucial for establishing a biologically plausible link between a drug and a disease. Our strategy is to explore longer paths to ensure no relevant biological context is missed, even if a simple one or two-hop path exists.
> While simpler methods may yield competitive metric scores, they are unsuitable for applications demanding transparency, human auditability, and context-rich biological reasoning. Our expanded evaluation confirms the competitive performance of our LLM-based reasoning approach while highlighting its unique strategic advantages in generalizability and interpretability.

---

> > ### Author Response · Authors · 2025-12-03
> >
> > **Weakness 3: Given the above, if simpler methods work better or similarly, then why should one consider using this approach instead of others for this drug repurposing task?**
> >
> > **Q3: In light of whatever the results of point 2 are, better justify how this approach fits in with other, recent computational drug repurposing approaches.**
> >
> > **Q4: Justify what role the LLM is playing here- why not simply conduct link prediction on a KG?**
> >
> > While link prediction methods are powerful, our research is fundamentally motivated by exploring the capabilities of Large Language Models (LLMs) for complex, interpretable reasoning in a biomedical context. Our primary goals were to determine:
> > * Feasibility of LLM-based Reasoning: Can an LLM, when supplied with structured KG context via Graph-of-Thoughts (GoT), effectively perform the drug repurposing task?
> > * Interpretability and Transparency: LLM-based reasoning, unlike many black-box LP models, intrinsically provides transparent reasoning traces. The GoT framework allows us to review the specific logical steps and retrieved biomedical pathways that led to the final prediction. This traceability is critical in clinical and biomedical applications, enabling human experts to audit and trust the system's conclusions.
> > * Human-in-the-Loop Potential: The natural language output from the LLM is conducive to human-in-the-loop intervention. Researchers can easily monitor the reasoning process and potentially inject novel, cutting-edge findings (e.g., new experimental results not yet in the KG) directly into the LLM's thought process to refine the outcome.
> > We believe these revisions clarify the methodological choices and strengthen the empirical validation of our work.

---

### Official Review · Reviewer_JvoB · 2025-10-28

**Soundness:** 1
**Presentation:** 2
**Contribution:** 1
**Rating:** 0
**Confidence:** 4

**Summary:**

This paper applied graph of thoughts method for drug repurposing for rare diseases. It utilizes PrimeKG as the base knowledge graph, and works with LLM through GoT to generate cypher queries to search PrimeKG and summarize/evaluate the outputs. It has been tested on 90 pairs of drugs and diseases on public data and private data.

**Strengths:**

- adapting graph of thoughts to drug repurposing for rare diseases
- evaluate the proposed methods with self-developed baseline method biomni for evaluation.
- Based on the existing large scale knowledge graph for drug discovery PrimeKG.

**Weaknesses:**

- This paper lacks algorithm novelty. it is an application of existing methods, such as graph of thoughts, neo4j search using cypher queries, LLM to summarize results. It is more toward a line of engineering work to build a tool.
- The evaluation is vague, It is rather a hypothesis generating and evaluation. It does not compare with the related work or baselines, such as compare with tree of thoughts, or chain of thoughts methods, or compare directly with the strong reasoning models.
- the evaluation is based on one public dataset and one private dataset, but it is not clear why these matter, as the main knowledge database is still PrimeKG.

**Questions:**

- can you please compare with more related baselines, for example, just for the reasoning part, you can compare your graph of thoughts with tree of thoughts and chain of thoughts.
- can you just use the powerful reasoning models (such as GPT-5) to directly address the questions and compare the reasoning steps with your graph of thoughts
- There are different knowledge graphs for drug repurposing (https://academic.oup.com/bib/article/25/6/bbae461/7774899), can you please justify that PrimeKG is the best KG for your application.

---

> ### Author Response · Authors · 2025-12-03
>
> We sincerely thank the reviewer for their careful reading of our manuscript and for providing valuable feedback. We have revised our paper based on your comments and hope that the following clarifications address your concerns.
>
> **Weakness 1: This paper lacks algorithm novelty. it is an application of existing methods, such as graph of thoughts, neo4j search using cypher queries, LLM to summarize results. It is more toward a line of engineering work to build a tool.**
>
> 1. Algorithmic Novelty
>
> We appreciate the comment regarding algorithmic novelty. Our paper’s primary research objective is to evaluate the efficacy of the Graph-of-Thoughts (GoT) reasoning framework when applied to the specialized domain of drug repurposing using structured knowledge. Therefore, our focus is on the application and evaluation of this combined system, rather than the development of a novel LLM-centric algorithm.
>
> **Weaknesses 2: The evaluation is vague, It is rather a hypothesis generating and evaluation. It does not compare with the related work or baselines, such as compare with tree of thoughts, or chain of thoughts methods, or compare directly with the strong reasoning models.**
>
> **Q1: can you please compare with more related baselines, for example, just for the reasoning part, you can compare your graph of thoughts with tree of thoughts and chain of thoughts.**
>
> **Q2: can you just use the powerful reasoning models (such as GPT-5) to directly address the questions and compare the reasoning steps with your graph of thoughts**
>
> 2. Vague Evaluation and Benchmarking
>
> We agree that a more comprehensive benchmark was necessary. We have updated Table 1 and the corresponding Results section (starting on Line 324 in the revised manuscript) to include further benchmarking against state-of-the-art methods and relevant baselines, addressing this concern regarding the evaluation scope.
>
> **Weakness 3: the evaluation is based on one public dataset and one private dataset, but it is not clear why these matter, as the main knowledge database is still PrimeKG.**
>
> 3. Clarity on Datasets: PrimeKG vs. Evaluation Sets
>
> The reviewer raises a point about the need to describe the role of the knowledge graph (PrimeKG) and the role of the evaluation datasets. They serve two fundamentally different purposes:
> PrimeKG serves as the underlying Knowledge Graph (KG) for the pathfinding step. It is not an evaluation dataset. We use PrimeKG to generate diverse biologically meaningful paths connecting the input drug and disease. This enables the model to reason over different biological pathways and possibilities. These paths involve intermediate entities such as genes, proteins, biological pathways, and side effects. For example, a retrieved path might look like:
> ('Drug A', 'binds to', 'Target B'), ('Target B', 'interacts with', 'Gene C'), ('Gene C', 'is associated with', 'Disease D')
> These retrieved paths are then passed to the LLM as context for the GoT reasoning process. The evaluation datasets consist of held-out drug-disease pairs along with their ground-truth labels (e.g., whether the drug is an approved indication for the disease). They provide the input pairs that are fed into the entire pipeline (Input Processing, KG Pathfinding, GoT Reasoning). The final prediction from the LLM is then compared against the ground truth labels from these datasets to calculate performance metrics (e.g., accuracy).
> PrimeKG provides the knowledge needed for reasoning, while the evaluation datasets provide the test samples for scoring accuracy. Both are distinct and necessary components for a comprehensive evaluation of our system.
>
> **Q3: There are different knowledge graphs for drug repurposing (https://academic.oup.com/bib/article/25/6/bbae461/7774899), can you please justify that PrimeKG is the best KG for your application.**
>
> 4. Selection of PrimeKG
>
> We stand by the choice of PrimeKG as the most suitable knowledge graph for this work due to several key advantages:
> * PrimeKG is intentionally designed to be easily extensible, allowing researchers to readily incorporate new biological knowledge as it is discovered, which is critical in a fast-moving field like drug discovery.
> * Richness: The KG's wide variety of node types and substantial quantity of nodes and edges enable the generation of diverse and in-depth paths, facilitating sophisticated LLM reasoning.
> * Consistency for Evaluation: As a static KG last updated in 2022, PrimeKG provides a consistent and stable knowledge base, which is vital for a fair and reproducible evaluation of drug-disease pairs.

---

### Meta-Review · Area_Chair_Q6uo · 2025-12-02

**Summary:**

All of the reviews of this paper are negative. We suggest authors to revise their work before future submission.

**Reviewer Concerns:**

The authors did not provide any response.

**Reviewer Scores:**

All scores are negative.

---

### Decision · Program_Chairs · 2026-01-26

Reject